# Selective oxidation of methane to $C_{2+}$ products over Au-CeO$_2$ by photon-phonon co-driven catalysis

Chao Wang[1], Youxun Xu [1], Lunqiao Xiong [1,2], Xiyi Li[1], Enqi Chen[1], Tina Jingyan Miao[1], Tianyu Zhang [3] ✉, Yang Lan [1] & Junwang Tang [1,2] ✉

Direct methane conversion to high-value chemicals under mild conditions is attractive yet challenging due to the inertness of methane and the high reactivity of valuable products. This work presents an efficient and selective strategy to achieve direct methane conversion through the oxidative coupling of methane over a visible-responsive Au-loaded CeO$_2$ by photon-phonon co-driven catalysis. A record-high ethane yield of 755 µmol h$^{-1}$ (15,100 µmol g$^{-1}$ h$^{-1}$) and selectivity of 93% are achieved under optimised reaction conditions, corresponding to an apparent quantum efficiency of 12% at 365 nm. Moreover, the high activity of the photocatalyst can be maintained for at least 120 h without noticeable decay. The pre-treatment of the catalyst at relatively high temperatures introduces oxygen vacancies, which improves oxygen adsorption and activation. Furthermore, Au, serving as a hole acceptor, facilitates charge separation, inhibits overoxidation and promotes the C-C coupling reaction. All these enhance photon efficiency and product yield.

Methane, the main component of natural gas, has been used as a low-cost and clean fuel for heating. With large reserves of natural gas being discovered, the use of methane as a potential primary raw material for chemical synthesis becomes attractive. Industrially, methane upgrade is realised via two steps: an initial reforming procedure to generate synthesis gas, and a subsequent Fischer-Tropsch process to convert the synthesis gas into long-chain hydrocarbons[1,2]. Compared with the two-step process, the direct conversion of methane into high-value chemicals is not only scientifically viable but also economically cost-effective. Up to now, tremendous effort has been made to directly convert methane into methanol or multi-carbon products (C$_{2+}$) via non-oxidative coupling of methane (NOCM), oxidative coupling of methane (OCM), and partial oxidation of methane[3-6]. However, due to the highly symmetrical molecular structure and high C-H bond energy, efficient and selective methane conversion always requires high temperatures (>600 °C) and/or expensive oxidants (e.g., H$_2$O$_2$ and H$_2$SO$_4$)[7-9]. In recent years, there has been a growing interest in the development of catalysts and reaction conditions for the direct conversion of methane under mild conditions. This is a challenging but promising area of research, with the potential to significantly reduce the cost and environmental impact of methane conversion.

Photocatalysis is an attractive approach to drive chemical reactions, and offers a promising solution for direct methane conversion. Instead of using heat, photocatalysis employs photons to drive reactions under mild conditions. Recent studies have shown that methane can be photocatalytically converted into mono-carbon (C$_1$) oxygenates (e.g., CH$_3$OH, CH$_3$OOH, HCHO, etc.) by metal oxides loaded with co-catalysts in batch reactors[10-14]. Conversion of methane into C$_{2+}$ chemicals is economically more profitable, but also technically more challenging. Nevertheless, photocatalytic methane upgrade in the absence of oxidants (NOCM) shows the potential to achieve C$_{2+}$ products with a selectivity of ca. 95% using catalysts such as ZnO, TiO$_2$ and Ga$_2$O$_3$, but the yield is very low (ca. 2 µmol h$^{-1}$) due to the endothermic nature of the reaction[15-17]. For photocatalytic OCM, the presence of oxygen promotes the yield of C$_{2+}$ in methane conversion. Our study has demonstrated the continuous conversion of methane into ethane

[1]Department of Chemical Engineering, University College London, London WC1E 7JE, UK. [2]Industrial Catalysis Center, Department of Chemical Engineering, Tsinghua University, Beijing 100084, China. [3]Beijing Key Lab for Source Control Technology of Water Pollution, College of Environmental Science and Engineering, Beijing Forestry University, Beijing 100083, P. R. China. ✉e-mail: tzhang@bjfu.edu.cn; jwtang@tsinghua.edu.cn

and ethylene using a Pt-CuO$_x$/TiO$_2$ photocatalyst in the presence of oxygen[18]. An improved ethane production rate of 6.8 μmol h$^{-1}$ with a moderate C$_2$ selectivity of 60% was obtained. Later, a Ag-AgBr/TiO$_2$ photocatalyst was developed, which increased the ethane yield to 35 μmol h$^{-1}$ with a C$_{2+}$ selectivity of 79% in a pressurised flow reaction system[19]. Additionally, a ZnO/TiO$_2$ hybrid photocatalyst loaded with Au was reported to selectively produce ethane at a yield of 100 μmol h$^{-1}$, achieving a selectivity of 90%[20]. Very recently, PdCu alloy and Au, working as hole-acceptors, were loaded onto UV-responsive TiO$_2$ to reduce the overoxidation process in photocatalytic OCM. A series of spectroscopic measurements were adopted to investigate the charge migration process[21,22]. It is worth noting that the photocatalysts in these systems are mainly TiO$_2$-based semiconductors, which are only active under UV light and show poor capability for visible light harvesting. On the other hand, overoxidation leads to low selectivity towards C$_{2+}$ chemicals, given that all products or intermediates generated in OCM are more reactive than methane itself. Thus, it is desirable to design photocatalysts that display excellent light harvesting and charge separation, efficient methane activation and effective co-catalysts to restrain overoxidation in methane conversion.

Ceria (CeO$_2$), a rare earth metal oxide, has displayed great potential in thermocatalysis[23]. CeO$_2$ has been used as a powerful oxidation catalyst in various reactions, including CO oxidation, NO$_x$ removal, organics oxidation, etc., due to its high oxygen mobility and the multi-valence nature of Ce[24–26]. For instance, Pd@CeO$_2$ supported on Al$_2$O$_3$ showed active and stable methane oxidation performance over a wide range of temperatures from 250 to 850 °C and was capable of completely converting methane into CO$_2$ at 400 °C[27]. Additionally, CeO$_2$ is a visible-light responsive semiconductor with an absorption edge of 400–500 nm depending on the size and morphology, making it a suitable candidate for visible-driven photocatalysis in applications such as environmental treatment[28,29]. For instance, CeO$_2$ doped with

Ru and modified by Au co-catalyst was used for CO$_2$ methanation[30]. A one-pass CO$_2$ conversion of 75% and a methane selectivity of 100% was achieved at a GHSV of 80,000 mL g$^{-1}$ h$^{-1}$ under visible light irradiation. Recently, an intertwined network of fibrous Rh#CeO$_2$ composite was reported for the photocatalytic reforming of methane with carbon dioxide[31]. The high yield and selectivity originated from the efficient partitioning of charges by Rh modification. With careful catalyst design, CeO$_2$ could harvest both UV and visible photons and serve as a high-efficiency photocatalyst for methane oxidation.

In this study, we designed and synthesised CeO$_2$ photocatalyst modified with Au (Au-CeO$_2$−300) for photocatalytic oxidation of methane. An efficient and stable ethane production at a yield of 755 μmol h$^{-1}$ (15,100 μmol g$^{-1}$ h$^{-1}$) with a high C$_{2+}$ selectivity of 98% has been achieved, which is also attractively stable (>120 h). It is found that pretreating the Au-CeO$_2$ catalyst at 300 °C introduces a certain level of defects (e.g., Ce$^{3+}$ and oxygen vacancies) in CeO$_2$, which is beneficial for methane oxidation. Finally, the role of Au co-catalyst in charge separation, overoxidation and C-C coupling is thoroughly discussed.

## Results and discussion

The photocatalytic methane oxidation was tested in a pressurised flow reaction system[19]. No products are observed in the absence of light, catalyst, or methane (Supplementary Fig. 1). Upon irradiation under 365 nm LED, Au-CeO$_2$−300 shows the capability to convert methane to C$_2$ to C$_4$ products (C$_{2+}$, including C$_2$H$_6$, C$_2$H$_4$, C$_3$H$_8$, C$_3$H$_6$ and C$_4$H$_{10}$) and a small amount of CO$_2$ in the presence of air (Fig. 1a, Supplementary Fig. 1 and Supplementary Table 1). When using pristine CeO$_2$ as the photocatalyst, CO$_2$ is detected as the main product at a yield of 478 μmol h$^{-1}$ with the co-production of C$_{2+}$ hydrocarbons. The selectivity of C$_{2+}$ products is only 13% for CeO$_2$ (Fig. 1a). A variety of metals and metal oxides were loaded onto CeO$_2$ to investigate the effect of co-catalysts on the performance of methane oxidation. Pt, Ru, Pd and Rh

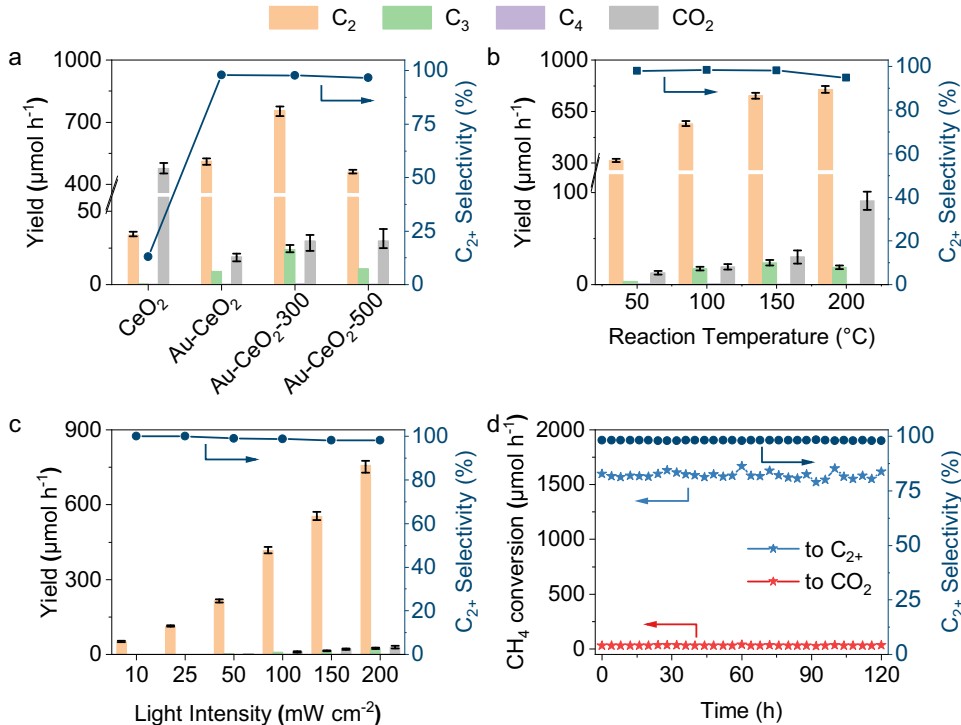

**Fig. 1 | Photocatalytic oxidation of methane. a** Product yields and C$_{2+}$ selectivity over CeO$_2$, Au-CeO$_2$, Au-CeO$_2$-300 and Au-CeO$_2$−500; **b** Product yields and C$_{2+}$ selectivity over Au-CeO$_2$−300 operated at different temperatures; **c** Products yield and C$_{2+}$ selectivity over Au-CeO$_2$-300 under different light intensities; **d** Long-term test of methane conversion rate and C$_{2+}$ selectivity over Au-CeO$_2$-300. Error bars represent standard deviations calculated from the performance tests of the photocatalysts prepared in three different batches. Unless otherwise specified, the reaction conditions applied are 50 mg catalyst, methane to air = 200:1, GHSV = 480,000 mL h$^{-1}$ g$^{-1}$, Pressure = 5 bar, Temperature = 150 °C, 365 nm LED, light intensity = 200 mW cm$^{-2}$.

dramatically improve the oxidation of $CH_4$ to $CO_2$ (Supplementary Fig. 2). The oxide co-catalysts, such as $CoO_x$, $MnO_x$, and $CuO_x$ display improved $C_{2+}$ selectivity and suppressed overoxidation (Supplementary Fig. 3). Among all the co-catalysts used, Au, when loaded onto $CeO_2$, displays the highest yield and selectivity towards $C_{2+}$ hydrocarbons, especially $C_2H_6$ (Supplementary Figs. 3 and 4). The production rate of $CO_2$ is reduced from 478 to 30 $\mu mol\ h^{-1}$. A $C_2H_6$ yield of 512 $\mu mol\ h^{-1}$ with a high $C_{2+}$ selectivity of 98% is obtained. Pre-treating Au-$CeO_2$ at 300 °C in air leads to an increase in the yield of all products. The yield of $C_2H_6$ is further improved to 755 $\mu mol\ h^{-1}$, while the high $C_{2+}$ selectivity of 98% is not affected. The isotopic labelling experiments unambiguously prove that the produced $C_{2+}$ and $CO_2$ are from $CH_4$ and $O_2$ in the reactant (Supplementary Fig. 5).

The loading amount of Au on $CeO_2$ was firstly optimised. As discussed above, the major product obtained over pure $CeO_2$ was $CO_2$, suggesting an intense overoxidation process. The product selectivity is shifted towards $C_{2+}$ hydrocarbons from 13% to 79% even with 0.1 wt.% Au loading (Supplementary Fig. 6). The yield of $C_2H_6$ increases from 427 to 755 $\mu mol\ h^{-1}$ as the Au loading increases from 0.1 wt.% to 1 wt.%. Further increasing the Au loading causes a decrease in the yield while an increase in the selectivity of $C_{2+}$ products. This suggests that the co-catalyst Au plays an important role in either promoting C-C coupling or suppressing overoxidation in the methane oxidation process. However, excessive Au covers the surface of $CeO_2$ and blocks light absorption, resulting in decreased photon efficiency. Pretreating Au-$CeO_2$ at elevated temperatures can have a positive impact on the performance of the catalyst. Au-$CeO_2$ was pre-treated in a muffles furnace from 200 to 500 °C. The optimised temperature for pretreatment is 300 °C (catalyst denoted Au-$CeO_2$-300, Supplementary Fig. 7). High temperatures such as 500 °C (Au-$CeO_2$-500) cause a decrease in the yield of $C_2H_6$ from 755 to 459 $\mu mol\ h^{-1}$. The actual Au amount in Au-$CeO_2$, Au-$CeO_2$-300, and Au-$CeO_2$-500 are 0.83 wt.%, 0.85 wt.%, and 0.81 wt.%, respectively, as measured by inductively coupled plasma-atomic emission spectrometry.

To optimise the reaction conditions for the photocatalytic conversion of methane using Au-$CeO_2$-300, various parameters are studied. The effect of the reaction pressure on the activity of the catalyst was firstly studied. Increasing the reaction pressure from 1 bar to 5 bar leads to an increase in the yield of all products (Supplementary Fig. 8). The yield of $C_2H_6$ increases from 447 to 755 $\mu mol\ h^{-1}$, while the selectivity towards $C_{2+}$ is hardly affected. The product yields under 6 and 7 bar are similar to that achieved under 5 bar. Thus, 5 bar is selected as the optimised pressure. Next, the methane-to-air ratio is studied at a constant gas hourly space velocity (GHSV) of 480,000 mL $h^{-1}\ g^{-1}$ using Ar as a balance. When $O_2$ is absent in the reaction atmosphere, a low $C_2H_6$ yield of 71 $\mu mol\ h^{-1}$ is obtained (Supplementary Fig. 9). As the concentration of air in the reactant increases, the yields of all products are improved. This indicates that oxygen gas can have a positive effect on photocatalytic methane conversion. However, the yield of $CO_2$ surges from 30 to 124 $\mu mol\ h^{-1}$ when the methane-to-air ratio changes from 200:1 to 12:1 with the $C_2H_6$ yield slightly increased from 755 to 859 $\mu mol\ h^{-1}$. Overall, $O_2$ is indispensable in photocatalytic OCM, however, too much $O_2$ causes overoxidation of $CH_4$ to $CO_2$. To achieve both high yield and high selectivity of the high-value $C_{2+}$ products, the methane-to-air ratio of 200:1 is chosen for subsequent studies. Another key factor is the reaction temperature, which would likely promote photocatalysis, i.e. the concept of photon-phonon co-driven catalysis. Au-$CeO_2$-300 was then tested at temperatures ranging from 50 to 200 °C to investigate the coupling effect of thermal catalysis with photocatalysis (Fig. 1b). When the reaction temperature increases from 50 to 150 °C, the yields of $C_2H_6$ more than doubles, increasing from 317 to 755 $\mu mol\ h^{-1}$ while the selectivity to $C_{2+}$ hydrocarbons remains nearly identical at 98%. $CO_2$ production also nearly doubles. The yield of $C_2H_6$ at 200 °C slightly increases to 792 $\mu mol\ h^{-1}$, while the $CO_2$ production is more than tripled (91 $\mu mol\ h^{-1}$) compared with that at 150 °C. Thus, the

optimised reaction condition for the photon-phonon co-driven catalysis is 150 °C and 5 bar. The product yield of Au-$CeO_2$-300 at higher temperatures from 150 to 200 °C was also measured in dark (Supplementary Fig. 10). Only a trace amount of $CO_2$ is detected at 200 °C. Thus, it can be concluded that the selective catalytic conversion of methane into $C_{2+}$ products is not driven by heat only while the catalytic performance of Au-$CeO_2$-300 at different temperatures (Fig. 1b) again suggests coupling photocatalysis with thermocatalysis can dramatically enhance the yield of $C_{2+}$ products.

Moreover, the effect of GHSV was also studied. It is found that as the GHSV increases from 160,000 to 480,000 mL $h^{-1}\ g^{-1}$, the yield of all products and the selectivity towards $C_{2+}$ products increase (Supplementary Fig. 11). High GHSV causes reduced detention time of reactants on the catalyst bed, which is beneficial for the suppression of overoxidation but decreases overall methane conversion. Furthermore, the photocatalytic methane oxidation was measured under various light intensities from 10 to 200 mW $cm^{-2}$. The product yield is approximately in a linear relationship with the light intensity (Fig. 1c), indicating that photo-generated charge carriers have the high utlisation efficiency. Since $CeO_2$ is a visible-light responsive photocatalyst, Au-$CeO_2$-300 was also tested under other light sources, including a Xe lamp (visible light, $\lambda > 420\ nm$) and a blue LED (450 nm), and a Xe lamp equipped with a $\lambda > 475\ nm$ and $\lambda > 550\ nm$ filter (Supplementary Fig. 12). As shown in Supplementary Fig. 13, the $C_2H_6$ yields under visible light and blue LED are 158 and 57 $\mu mol\ h^{-1}$, respectively, even higher than most of the reported results measured under strong UV light (Supplementary Table 2). The photocatalytic methane conversion performance of Au-$CeO_2$-300 follows the absorption spectrum of $CeO_2$ (Supplementary Figs. 11 and 12). Thus, the photo-generated charges by $CeO_2$ are the driving force for the conversion of methane to $C_{2+}$ products. No products are detected over Au-$TiO_2$ and Au-ZnO under >420 nm irradiation (Supplementary Fig. 14). This shows the superiority of $CeO_2$ as a visible-responsive photocatalyst and excludes the contribution of the plasmonic effect of Au to the activity under visible irradiation. Overall, the yield of $C_2H_6$, the main product, is as high as 755 $\mu mol\ h^{-1}$ and the selectivity towards $C_{2+}$ molecules is 98% under the optimised conditions. This performance is the record among all reported photocatalytic methane conversion reactions (Supplementary Table 2) and is >7 times higher than the recently reported benchmark photocatalyst Au-$TiO_2$/ZnO[20]. The apparent quantum efficiency of Au-$CeO_2$-300 at 365 nm is calculated to be 12%. In addition, the potential of Au-$CeO_2$-300 for oxidative coupling of methane under simulated solar irradiation was also measured using a Xe lamp equipped with an AM 1.5 filter (100 mW $cm^{-2}$) without external heating. The production rates of $C_2H_6$, $C_2H_4$, $C_3H_8$, and $C_4H_{10}$ are 104, 0.2, 4.5 and 51 $\mu mol\ h^{-1}$, respectively, suggesting the potential of the current catalyst for solar driven methane conversion although it is less productive than the above photon-phonon co-driven catalysis.

The durability and stability of Au-$CeO_2$-300 were assessed by conducting long-term tests under the optimised reaction conditions for 120 h (Fig. 1d). Methane is continuously converted into $C_{2+}$ products at a rate ranging from 1521 to 1665 $\mu mol\ h^{-1}$ with the $C_{2+}$ selectivity kept at 98%. The Au-$CeO_2$-300 photocatalyst after running for 120 h was then characterised by X-ray diffraction (XRD), transmission electron microscopy (TEM), and ultraviolet-visible diffuse reflectance spectroscopy (UV-Vis DRS). The results indicate that the phase structure, morphology band structure of $CeO_2$, and the particle size of the co-catalyst Au remained unchanged in Au-$CeO_2$-300 even after 120 h of reaction (Supplementary Figs. 15–17). These findings prove that Au-$CeO_2$-300 is a robust and stable photocatalyst and can efficiently and selectively convert methane into $C_{2+}$ hydrocarbons for an extended period under the current reaction conditions.

A thorough characterisation of the catalysts used in this study was conducted. XRD analysis of $CeO_2$ shows that all peaks are attributed to the cubic fluorite phase (JCPDS # 34-0394) of ceria (Fig. 2a). After the

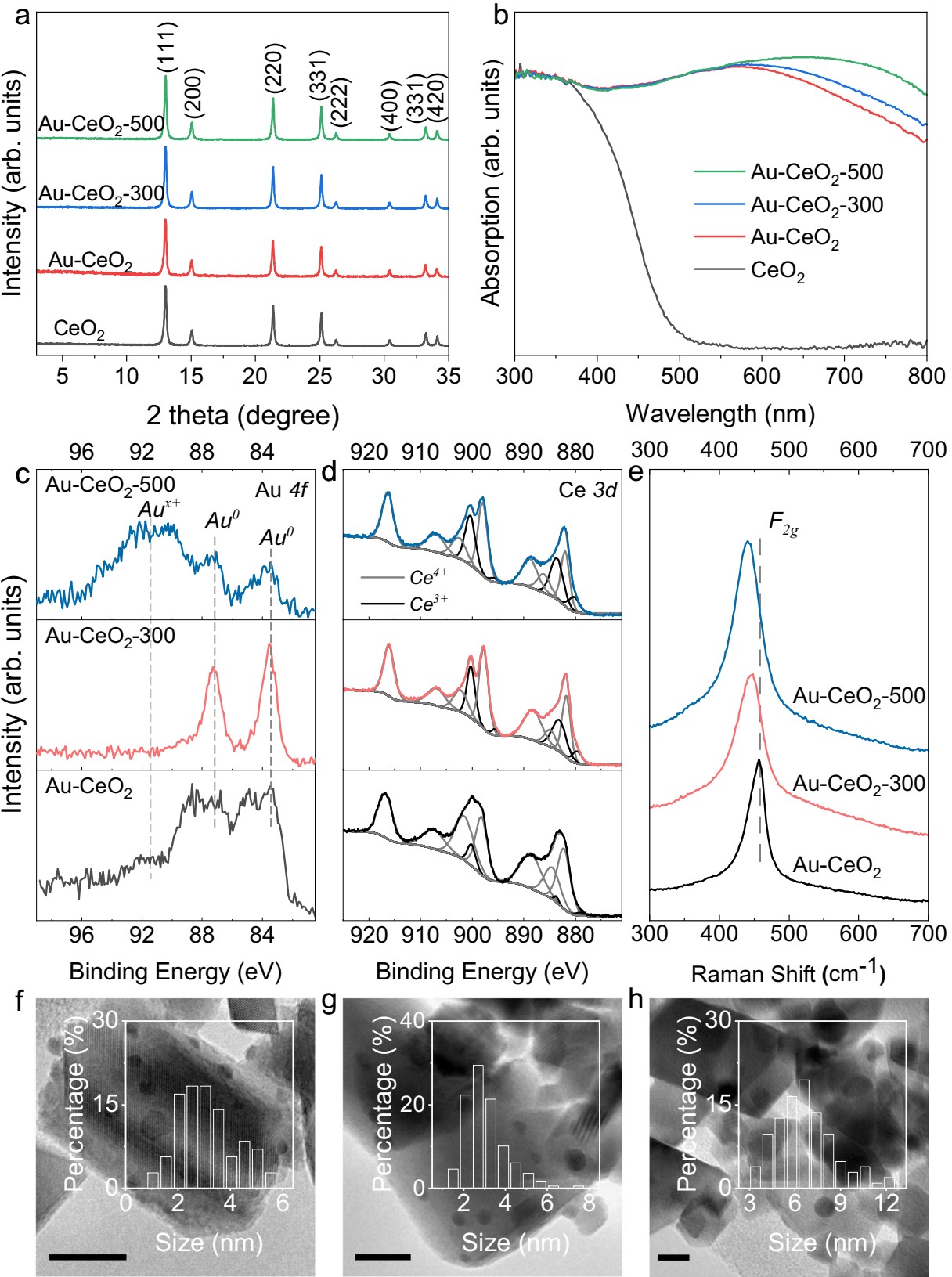

**Fig. 2 | Catalyst characterisation. a** XRD and (**b**) UV-Vis DRS spectra of CeO₂, Au-CeO₂, Au-CeO₂-300 and Au-CeO₂−500; **c** Au *4f*, (**d**) Ce *3d* XPS and (**e**) Raman spectra of Au-CeO₂, Au-CeO₂−300 and Au-CeO₂-500; TEM images of (**f**) Au-CeO₂, (**g**) Au-CeO₂-300 and (**h**) Au-CeO₂-500, insets show the size distribution of Au particles, scale bar: 10 nm.

loading of Au, no additional peaks for Au are detected, likely due to the small particle size and/or high dispersion of the co-catalyst. Similar diffraction features are observed for Au-CeO$_2$ after calcination at 300 °C and 500 °C, indicating that the lattice structure of the photocatalyst remains unchanged after the pre-treatment. The UV-Vis DRS spectrum of CeO$_2$ shows an absorption edge of 495 nm, corresponding to an optical bandgap of 2.5 eV (Fig. 2b), which is slightly smaller than the reported probably due to partially reduced Ce$^{4+}$ in the catalyst[32]. When 1 wt. % of Au is loaded on CeO$_2$, a strong absorption is observed across the whole visible range. The absorption in UV and blue regions (300-450 nm) is mainly contributed by CeO$_2$. The band at 600 nm is ascribed to the plasmonic effect of metallic Au nanoparticles on the surface of CeO$_2$[20]. Due to the closely packed Au nanoparticles on the CeO$_2$ support, a strong scattering in the visible range is observed[33]. When the Au loading amount is reduced to 0.1 wt.%, where the scattering is reduced, two absorption features originating from CeO$_2$ and plasmonic Au are clearly observed (Supplementary Fig. 18). The strong visible absorption is also reflected by the dark colour of Au-CeO$_2$ (Supplementary Fig. 19). A red shift of the plasmonic absorption is found after Au-CeO$_2$ is calcined at higher temperatures, particularly for Au-CeO$_2$-500, indicating that the pre-treatment at an extra high temperature may have caused growth or aggregation of the co-catalyst Au[34].

X-ray Photoelectron Spectroscopy (XPS) was performed to investigate the relation between the chemical states and the performance of photocatalysts. High-resolution Au 4$f$ spectra suggest that metallic Au is the main species of Au in Au-CeO$_2$ and Au-CeO$_2$-300 (Fig. 2c), while a large proportion of oxidised Au is found in Au-CeO$_2$-500. This indicates that the calcination of Au-CeO$_2$ at 500 °C in air caused the oxidation of Au into Au$_2$O$_3$[35]. The Ce 3$d$ spectra show that Ce$^{4+}$ and Ce$^{3+}$ co-exist in CeO$_2$, Au-CeO$_2$, Au-CeO$_2$-300 and Au-CeO$_2$-500 (Supplementary Fig. 20 and Fig. 2d)[36]. The bands in the Ce 3$d$ XPS spectrum of Au-CeO$_2$-300 display a clear shift towards lower binding energies compared to that of Au-CeO$_2$, which indicates a partial reduction of Ce$^{4+}$ to Ce$^{3+}$ (Supplementary Fig. 21). The proportion of Ce$^{3+}$ and Ce$^{4+}$ in the catalysts is calculated based on the integrated area of the corresponding bands and summarised in Supplementary Table 4. It shows that the Ce$^{3+}$ contents in CeO$_2$ and Au-CeO$_2$ are relatively low, 8.5% and 4.8%, respectively. After calcination at 300 °C, 24.6% of Ce$^{3+}$ is found in Au-CeO$_2$-300. The Ce$^{3+}$ content increases to 28.9% when the pre-treatment temperature reaches 500 °C. EPR was used to further investigate the valence change of cerium after the pre-treatment of Au-CeO$_2$ (Supplementary Fig. 22). The strongest band with a g value of 1.96 originates from the Ce$^{3+}$-O$^-$-Ce$^{4+}$ sites in CeO$_2$[37]. The intensity of this band increases with the pre-treatment temperature, suggesting more Ce$^{4+}$ is reduced to Ce$^{3+}$ after pre-treating at higher temperatures. The signals at g values of 1.88, 1.93, 2.08 and 2.14 are also related to Ce$^{3+}$ in the samples[38].

The generation of Ce$^{3+}$ in CeO$_2$ is often accompanied by the formation of oxygen vacancies (O$_V$)[39]. Raman spectroscopy was used to probe the formation of O$_v$ in Au-CeO$_2$ after pre-treatment. The $F_{2g}$ mode in the Raman spectrum of CeO$_2$ originates from the symmetrical stretching vibration of O$^{2-}$ around Ce$^{4+}$ and is sensitive to defects such as O$_v$[40,41]. The F$_{2g}$ peak shifts from 457 cm$^{-1}$ for Au-CeO$_2$ to 448 cm$^{-1}$ for Au-CeO$_2$-300, and further to 441 cm$^{-1}$ for Au-CeO$_2$-500 (Fig. 2e). The redshift of $F_{2g}$ mode is related to the presence of O$_v$ in the material[40,42]. O$_V$ in CeO$_2$ has been reported to promote the adsorption and activation of oxygen gas in catalytic oxidation reactions[42]. To exclude the effect of Au particle change during the calcination on the adsorption of O$_2$, CeO$_2$ was calcined at 300 °C and 500 °C to obtain CeO$_2$-300 and CeO$_2$-500 and measured for oxygen adsorption. The results indicate that CeO$_2$-300 and CeO$_2$-500 display more oxygen uptake than CeO$_2$ (Supplementary Fig. 23). The O 1$s$ XPS spectra also reveal that after the pre-treatment, a large amount of chemically adsorbed oxygen-containing species are detected on the surface of the Au-loaded CeO$_2$ (Supplementary Fig. 24). The O$_2$$^-$ trapping EPR experiment indicates that the photocatalytic oxygen activation capability of CeO$_2$ is improved by the existence of O$_v$ (Supplementary Fig. 25). Improved photocatalytic methane oxidation is also observed for CeO$_2$-300 and CeO$_2$-500 compared with CeO$_2$ (Supplementary Fig. 26), in agreement with the Raman, O$_2$ adsorption and O$_2$$^-$ trapping EPR results. Pretreatment of Au-CeO$_2$ introduces O$_v$ into CeO$_2$, which promotes the adsorption and activation of O$_2$ and is beneficial for photocatalytic methane oxidation.

The TEM images of Au-CeO$_2$, Au-CeO$_2$-300 and Au-CeO$_2$-500 reveal that the Au nanoparticles are well-dispersed on CeO$_2$ (Fig. 2f–h). No significant change in the crystals of CeO$_2$ is observed after pretreatment (Supplementary Fig. 27). The Au size distribution indicates that the Au nanoparticles on Au-CeO$_2$ are between 1 and 6 nm, while >83% of Au on Au-CeO$_2$-300 displays particle size from 2 to 4 nm. It suggests that the Au nanoparticles smaller than 2 nm grow into sizes of 2–4 nm after pre-treatment at 300 °C. The slight growth of Au nanoparticles at 300 °C is verified by calcination Au-CeO$_2$-Na with smaller Au nanoparticles at 300 °C (Supplementary Figs. 28–30). The wide size distribution of Au in Au-CeO$_2$ also explains its widened Au 4$f$ XPS spectrum compared to that of Au-CeO$_2$-300 (Fig. 2c). However, increasing the calcination temperature to 500 °C leads to the intensive growth of Au nanoparticles to ~6 nm. The TEM images did not reveal any oxidised Au surface, which could be due to the reduction of the Au$_2$O$_3$ to the metallic state by the electron beam during TEM measurement.

Combining the results obtained from UV-Vis DRS, Raman, XPS, EPR, and TEM analysis, it can be concluded that pre-treating Au-CeO$_2$ at 300 °C induces a change of the catalyst, generating Ce$^{3+}$ sites and oxygen vacancies that improve oxygen adsorption and activation, and cause slight growth of the Au nanoparticles. However, it causes the intensive growth and oxidation of the co-catalyst Au when the pretreatment temperature is further increased to 500 °C, leading to reduced catalytic performance.

The yield and product selectivity of a photocatalytic system are often affected by two important factors, which are charge separation and surface reaction. Firstly, the charge separation and migration process of photocatalytic methane oxidation by CeO$_2$ and Au-CeO$_2$-300 was investigated by photo-induced absorption (PIA) spectroscopy. An in situ UV-Vis DRS system was developed to measure the reflectance of the photocatalyst in dark or under light irradiation in various reaction atmospheres (Supplementary Fig. 31)[43]. The PIA can be calculated by the following equation:

$$\Delta Abs = \frac{R_{dark} - R_{light}}{R_{dark}} \times 100\% \qquad (1)$$

Where R$_{dark}$ and R$_{light}$ are the reflectances measured in dark and under light conditions, respectively.

To ensure the reliability of the experimental setup and to eliminate the possibility of any interference from the irradiation source, a reference sample, BaSO$_4$ was tested under an Ar atmosphere. The reflectance spectra of BaSO$_4$ in dark and under light irradiation are found to overlap well with each other (Supplementary Fig. 32a). It suggests that no interference with the measurement is caused by the UV-LED as no PIA is detected for BaSO$_4$ (Supplementary Fig. 32b). The PIA spectrum of CeO$_2$ was subsequently observed under an Ar atmosphere. An evident difference between the reflectance of CeO$_2$ in dark and under light irradiation is observed (Supplementary Fig. 33a). The PIA spectrum of CeO$_2$ suggests that an absorption across the entire visible range is generated when CeO$_2$ is illuminated under a 365 nm LED (Supplementary Fig. 33b). Transient absorption studies reported that excited CeO$_2$ displays a positive absorption from 500 to 800 nm, which originates from the electron transition from the conduction band (CB) to the higher energy levels close to the CB (Supplementary

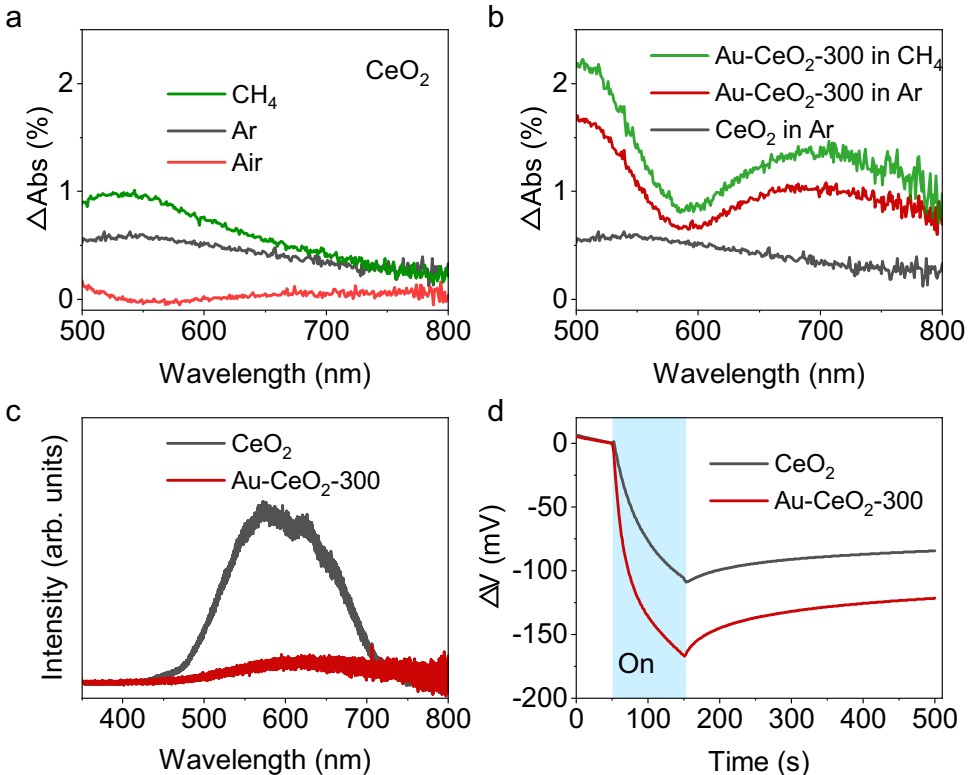

**Fig. 3 | Charge separation and migration. a** PIA spectra of $CeO_2$ in Ar, air and methane; **b** PIA spectra of $CeO_2$ in Ar, Au-$CeO_2$-300 in Ar and Au-$CeO_2$-300 in methane; **c** PL spectra and (**d**) OCVD plots of $CeO_2$ and Au-$CeO_2$-300.

Fig. 34a)[44]. A series of light on-off measurements of reflectance at 500, 600 and 700 nm show the repeatable decrease in the reflectance of $CeO_2$ upon irradiation, indicating the PIA signals are highly reproducible (Supplementary Fig. 35). The PIA spectra of $CeO_2$ are further investigated in air and methane atmospheres (Fig. 3a). The PIA of $CeO_2$ is quenched in the air atmosphere. Oxygen gas in air is a powerful electron scavenger and can be reduced by CB electrons of $CeO_2$[45,46]. In other words, most CB electrons from $CeO_2$ are consumed by oxygen gas when air is introduced, resulting in the reduced charge transition between the CB of $CeO_2$ and its closer energy levels (Supplementary Fig. 34b). On the other hand, in the methane atmosphere, an increase in the PIA spectrum is observed. This is ascribed to the fact that the function of methane is reversed to oxygen gas, ie. $CH_4$ is a photohole acceptor, which leads to improved charge separation and abundant photoelectrons at the CB of $CeO_2$ (Supplementary Fig. 34c). Consequently, the enhanced charge transition related to CB electrons is detected. XPS VB spectrum indicates $CeO_2$ display a VB potential of 2.2 V vs. NHE (Supplementary Fig. 36), which is sufficient for methane activation[47]. Considering the bandgap of $CeO_2$ is 2.5 eV, the CB potential is calculated to be -0.3 V vs. NHE, which is more negative than that for the oxygen reduction reaction (-0.16 V vs. NHE)[48]. Therefore, it can be concluded that during photocatalytic methane oxidation by $CeO_2$, electrons and holes are generated upon light irradiation, and then are consumed by oxygen gas and methane, respectively.

Subsequently, the PIA spectrum of Au-$CeO_2$-300 was measured in Ar to study the role of Au in charge separation and migration. The PIA spectrum of Au-$CeO_2$-300 exhibits two distinct features (Fig. 3b). Firstly, a significant increase in the PIA is detected over Au-$CeO_2$-300 compared with $CeO_2$, suggesting that Au plays a similar role as methane when $CeO_2$ is excited under UV irradiation. In other words, when Au-$CeO_2$-300 is irradiated under UV, the photoholes transfer from the VB of $CeO_2$ to Au and Au serves as a hole acceptor in Au-$CeO_2$-300. Secondly, a trough at ca. 600 nm in the PIA spectrum is observed,

which is caused by the ground-state bleaching of plasmonic Au[49,50]. A further increment in PIA is observed when Au-$CeO_2$-300 is measured in methane. This indicates that the oxidation potential of photo-holes on Au is still capable of methane activation. With the co-existence of Au and methane, more CB electrons in $CeO_2$ could be promoted to higher levels by the probe source. Therefore, the highest PIA is observed. To prove the reliability of this result, Pt, which is well-documented as an electron acceptor and co-catalyst in photocatalysis, was loaded on $CeO_2$ and measured for PIA (Supplementary Fig. 37). After loading Pt onto $CeO_2$, the PIA signal is quenched, indicating again that the PIA observed is attributed to photoelectrons in $CeO_2$ as the photo-generated electrons transfer from the CB of $CeO_2$ to Pt. This result confirms the conclusion above that Au works as a hole acceptor. The opposite results in the product distribution of methane conversion by Pt- or Au-modified $CeO_2$ also suggest that Pt and Au play different roles in photocatalytic methane oxidation (Supplementary Fig. 38). In situ Au 4$f$ XPS spectra of Au-$CeO_2$-300 were measured to further probe the function of Au in charge separation (Supplementary Fig. 39). A positive shift of 0.3 eV is observed for the Au 4$f$ XPS bands of Au-$CeO_2$-300 under light irradiation compared with that in dark. This suggests that under light irradiation, the transfer of holes from $CeO_2$ to Au is detected.

Density functional theory (DFT) calculations are also performed to simulate the charge migration process. Au-$CeO_2$-300 is simulated by loading a $Au_9$ cluster onto the (111) surface of $CeO_2$ with $Ce^{3+}$ and $O_V$s (Supplementary Figs. 40 and 41). The Density of States results show that Au 5d orbitals partially overlap with O 2p in Au-$CeO_2$-300, suggesting that Au possibly works as an electron donor (Supplementary Fig. 42). The charge density difference clearly indicates the decrease in the electron density of Au and the increase in the electron density of $CeO_2$ (Supplementary Fig. 43a), which means an electron transfer process from Au to $CeO_2$. This has provided direct theoretical evidence that Au works as a hole acceptor in Au-$CeO_2$-300. Similar

charge behaviour is also observed over Au₉-CeO₂ in the absence of $O_V$ (Supplementary Fig. 43b). The results obtained from the DFT calculation are in good agreement with the experimental analysis.

To gain further insights into the role of Au in charge separation, photoluminescence (PL) spectroscopy was conducted using a 325 nm UV laser as the excitation source. $CeO_2$ exhibits a strong fluorescence centred at ca. 600 nm (Fig. 3c). In contrast, the PL spectrum of Au-CeO₂-300 is quenched, indicating that the radiative charge recombination is significantly reduced after the loading of Au on $CeO_2$. A red shift of the PL spectrum is observed after Au-CeO₂ is pre-treated at 300 and 500 °C (Supplementary Fig. 44). This likely results from the recombinations related to the interband levels caused by the defects (e.g., $Ce^{3+}$ and $O_V$s) formed during pre-treatment, which is consistent with the XPS analysis. Electrochemical open circuit voltage decay (OCVD) is another widely used method to study the charge separation efficiency of semiconductors[51]. When irradiated by a Xe lamp, a negative photovoltage is observed over both $CeO_2$ and Au-CeO₂-300 (Fig. 3d), which arises from the separation of electrons and holes in the semiconductor electrodes. The accumulated photovoltage values on $CeO_2$ and Au-CeO₂-300 after irradiation for 100 s are 109 and 151 mV, respectively, indicating that Au can enhance the separation of photo-generated electrons and holes in $CeO_2$ by nearly 50%. Therefore, the following conclusions can be drawn from the charge separation and migration analysis. First, photo-generated electrons react with oxygen gas while holes activate methane during photocatalytic methane oxidation. Second, Au acts as a hole acceptor in Au-CeO₂-300 and can promote charge separation. Third, the photoholes transferred from the VB of $CeO_2$ to Au are sufficiently oxidative for methane activation.

After the charge separation and migration, the surface reactions take place, which involve reactant adsorption and activation, radical generation and transformation, product formation, and unexpected overoxidation[52]. The oxygen reduction reaction is an important half-reaction in all photocatalytic reactions with oxygen gas as the oxidant. The adsorption of $O_2$ on $CeO_2$ has been investigated and discussed (Supplementary Fig. 23). After adsorption on the catalyst surface, $O_2$ reacts with photoelectrons, resulting in the generation of surface superoxide radicals ($O_2^-$). Electron paramagnetic resonance (EPR) was thus used to monitor the production of $O_2^-$ radicals with 5,5-dimethyl −1-pyrroline N-oxide (DMPO) as the spin-trapping reagent. After irradiation for 30 s, an EPR signal with six distinct peaks is obtained (Fig. 4a), resulting from the coupling product of $O_2^-$ radicals and DMPO. Au-CeO₂-300 produces a much higher level of $O_2^-$ than $CeO_2$ because of its high charge separation capability. However, the electrochemical oxygen reduction LSV results indicate that Au does not directly improve the oxygen reduction capability of $CeO_2$, rather different from Pt (Supplementary Fig. 45). The photocatalytic performance of $CeO_2$ under an oxygen-rich environment displays a different trend in the product selectivity compared with Au modification, which also indicates that Au does not improve $O_2^-$ formation via working as an electron acceptor like Pt (Supplementary Fig. 46). The LSV and EPR measurements indicate that Au promotes charge separation by working as a hole acceptor, which is beneficial for methanol oxidation, and indirectly favourable for superoxide radical production from oxygen reduction. The formed superoxide radicals play two main roles in photocatalytic methane oxidative coupling. Firstly, it combines with protons generated from $CH_4$ oxidation half-reaction by photo-holes, to clear the surface of the catalyst for subsequent methane activation. Secondly, it reacts with the intermediates formed during methane oxidation, resulting in the production of $CO_2$. No EPR signals are detected for either $CeO_2$ or Au-CeO₂-300 in dark (Supplementary Fig. 47), indicating $O_2^-$ radicals are formed from the reaction of oxygen gas with photo-generated electrons. Oxygen exchange between the oxygen gas and the lattice oxygen of $CeO_2$ is also detected during photocatalytic methane oxidation by Au-CeO₂-300 (Supplementary Fig. 48), which is similar as the reported[20,53].

The other half reaction is the oxidation reaction of methane by photoholes. The adsorption of methane on $CeO_2$ and Au-CeO₂-300 was simulated using DFT calculations (Supplementary Fig. 49). The results show that although the adsorption of $CH_4$ on both $CeO_2$ and Au-CeO₂-300 surfaces is weak, which could be ascribed to the highly symmetric structure of the molecule, the adsorption energy of $CH_4$ on the Au-CeO₂-300 surface is 0.11 eV more negative than that on the $CeO_2$ surface, indicating that Au can slightly improve the adsorption of $CH_4$ on the $CeO_2$ surface, which is beneficial for the subsequent methane activation reaction. In situ DRIFTS reveal that the adsorption of methane can be improved by photo-irradiation (Supplementary Fig. 50). Activation of methane by photo-holes to generate $CH_3\cdot$ is observed by in situ DRIFTS in pure $CH_4$ (Supplementary Fig. 51). Under light irradiation, two bands at 2845 and 2880 cm$^{-1}$ are detected over both $CeO_2$ and Au-CeO₂-300, which are ascribed to the symmetric and asymmetric stretching vibration of C-H bond in $CH_3\cdot$ species[54]. The intensity of $CH_3\cdot$ in the spectrum of Au-CeO₂-300 is relatively stronger than that in $CeO_2$, indicating that Au could greatly promote the methane activation rate or $CH_3\cdot$ formation rate, which is also beneficial for the coupling of $CH_3\cdot$ to $C_2H_6$. DFT calculation also suggests that a reduced energy barrier is achieved for methane activation over Au-CeO₂-300 compared with $CeO_2$ (Supplementary Fig. 52). Another feature in the DRIFTS spectra is the band located at 2947 cm$^{-1}$, originating from the $CH_3O\cdot$ species (Supplementary Fig. 51)[55]. $CH_3\cdot$ is the first intermediate formed in the overoxidation process. The oxygen source in $CH_3O\cdot$ is the lattice oxygen of $CeO_2$ as the colour of both $CeO_2$ and Au-CeO₂−300 changes after the in situ DRIFTS measurement (Supplementary Fig. 53). The oxidation capability of the catalysts is also investigated by photo-electrochemical oxidation reaction. The solubility of methane is low in most electrolytes and the interaction of methane with the electrode surface is rather weak. Thus, photoelectrochemical water oxidation is used to investigate the oxidation capability of the photocatalysts, considering water oxidation requires a similar potential as methane activation[52]. Au-CeO₂-300 shows a higher photocurrent than $CeO_2$ across the entire potential window of the linear sweep voltammetry plot (Supplementary Fig. 54), indicating that more water molecules are activated and oxidised on Au-CeO₂-300 than $CeO_2$. The long-term transient photocurrent response of the two catalysts further confirms the stronger oxidation capability of Au-CeO₂-300 compared to $CeO_2$ (Fig. 4b).

To obtain an in-depth understanding of the photocatalytic methane oxidation process over $CeO_2$ and Au-CeO₂-300, in situ diffuse reflectance infrared Fourier transform spectroscopy (DRIFTS) was carried out. The DRIFTS spectra of the catalysts in dark in the reaction atmosphere (Supplementary Fig. 55) were used as the baseline when performing the measurement under light irradiation. As shown in Supplementary Fig. 56, the peak at 2885/2843 cm$^{-1}$ is associated with the C-H stretching vibration of the adsorbed $CH_3\cdot$ species[14,19,20,56], which is the first intermediate generated from methane activation by photoholes. Two peaks at 1402 and 1566 cm$^{-1}$, and three peaks at 1373, 1433 and 1548 cm$^{-1}$ are observed on the spectra of $CeO_2$ and Au-CeO₂-300, respectively. These peaks are attributed to HCOO· species, which is another important intermediate during the overoxidation process[55]. A carbonate species is detected at 1263/1236 cm$^{-1}$ (Supplementary Figs. 56b and 57)[19]. The overoxidation product $CO_2$ is observed at 2360 cm$^{-1}$ for both photocatalysts. The IR band positions of all detected species are summarised in Supplementary Table 5. A red shift in the bands of all adsorbed species (e.g., $CH_3\cdot$, $CO_3^{2-}\cdot$ and HCOO·) is observed in the spectrum of Au-CeO₂-300 compared with $CeO_2$, indicating different adsorption sites of the intermediates on the surface of the two catalysts. As discussed in the above charge separation section, Au acts as a hole acceptor in Au-CeO₂-300 and methane is activated by holes transferred from $CeO_2$ to Au. Thus, the intermediates formed in the subsequent oxidation steps are possibly adsorbed on Au or at the interface of Au and $CeO_2$.

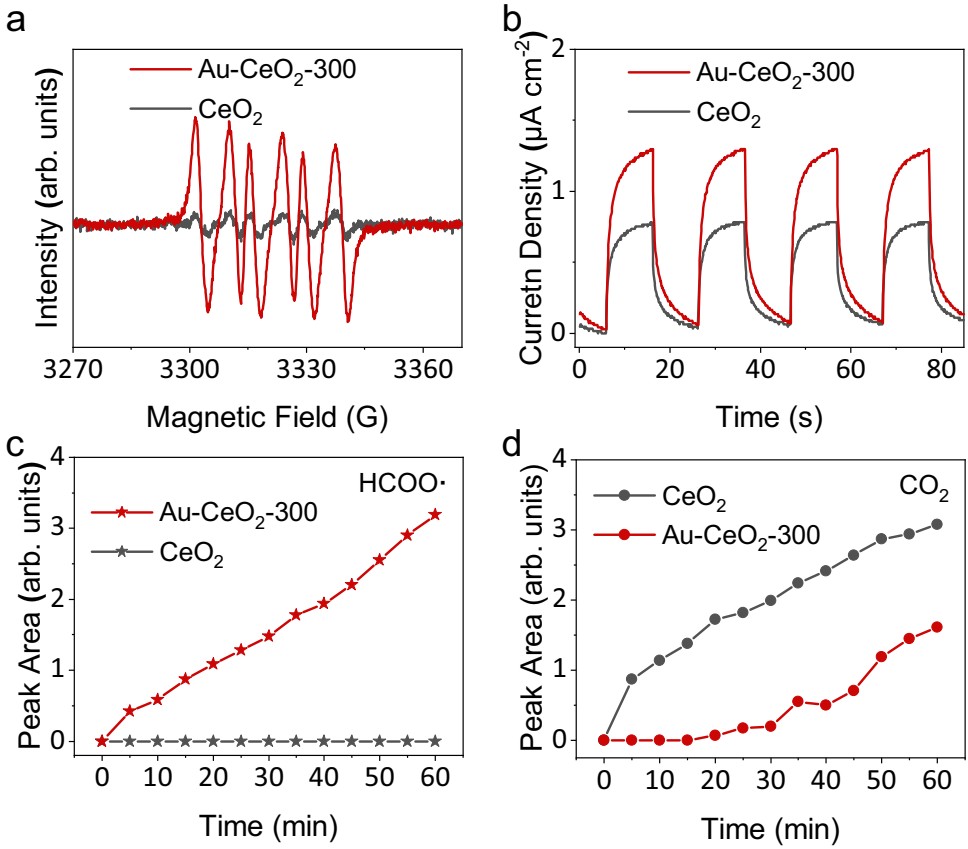

**Fig. 4 | Surface chemistry investigations. a** EPR spectra of $CeO_2$ and Au-$CeO_2$-300 for $O_2^-$ trapping; **b** Transient photocurrent response of $CeO_2$ and Au-$CeO_2$-300 at the bias potential of 0.3 V vs. Ag/AgCl; Evolution of (**c**) HCOO· species and (**d**) $CO_2$ during the in situ DRIFTS measurement of $CeO_2$ and Au-$CeO_2$-300 (methane to air = 200:1).

The quantity of survived species observed by DRIFTS is plotted as a function of reaction time to provide insights into the evolution and transformation of intermediates during photocatalytic methane oxidation (Fig. 4c, d and Supplementary Fig. 58). The amount of $CH_3$· first increases and then fluctuates at a certain concentration for both $CeO_2$ and Au-$CeO_2$-300 (Supplementary Fig. 58). However, the peak intensity of Au-$CeO_2$-300 is much stronger than that of $CeO_2$, indicating the superior methane activation capability of Au-$CeO_2$-300. The formed $CH_3$· follows two routes: (1) two $CH_3$· couple into $C_2H_6$, and (2) $CH_3$· is overoxidised, leading to the production of $CO_2$. Gas-phase $CH_3$· species has been observed using EPR and photoionisation mass spectroscopy[57,58]. Thus, the coupling of $CH_3$· could occur in the gas phase or on the catalyst surface. In the case of gas-phase radical reaction, Au can facilitate the desorption of $CH_3$· from the catalyst surface due to the lowest adsorption energy based on the adsorption energies of $CH_3$· calculated on the surface of $CeO_2$ and Au-$CeO_2$-300 (Supplementary Table 6). This suggests that $CH_3$· radicals on Au can readily desorb and couple into $C_2H_6$, while the $CH_3$· on $CeO_2$ likely undergoes overoxidation by photoholes or surface $O_2^-$ species since it is strongly bonded with O of $CeO_2$. The other case, where the radical reaction occurs on the surface of the catalyst, was also investigated by DFT simulation. The results indicate that the energy barrier for $CH_3$· coupling on the Au surface is much lower than that on the $CeO_2$ surface (Supplementary Fig. 59). This could also be revealed by the strong $CH_3O$· peaks observed in the DRIFTS spectrum of $CeO_2$ (Supplementary Fig. 51). Then, the strongest peak related to HCOO· is analysed (Fig. 4c). The peak area of this band keeps increasing with time over Au-$CeO_2$-300, indicating the accumulation of HCOO· on the surface of the catalyst. It suggests the consumption of HCOO· is much slower than its formation, and HCOO· oxidation is the rate-limiting step in methane oxidation to $CO_2$ over Au-$CeO_2$-300. In contrast, this band is

not detected when $CeO_2$ is used as the catalyst, and the intensity of the other two bands at 1402 and 1566 $cm^{-1}$ related to HCOO· is also weak. This implies that the HCOO· radicals formed on $CeO_2$ can be facilely converted to $CO_2$, the final overoxidation product. Thus, a very low amount of HCOO· is observed on the surface of $CeO_2$. This is also evidenced by the instant evolution of gaseous $CO_2$ over $CeO_2$ under photocatalytic reaction conditions (Fig. 4d), while only a weak signal of $CO_2$ is observed after reaction for 20 min when Au-$CeO_2$-300 is used as the catalyst. The DRIFTS analysis reveals that the loading of Au on $CeO_2$ could efficiently suppress the overoxidation reaction in photocatalytic methane oxidation by restricting the transformation of HCOO· radicals. To confirm this, the photocatalytic oxidation of HCOONa over $CeO_2$, Au-$CeO_2$-300, and Pt-$CeO_2$ was measured (Supplementary Fig. 60). The $CO_2$ production rate over $CeO_2$, Au-$CeO_2$-300, and Pt-$CeO_2$ are 457, 43, and 843 $\mu$mol $h^{-1}$, respectively. The results reveal that the oxidation of HCOO· to $CO_2$ is restricted by loading Au onto $CeO_2$. However, the $CO_2$ production rate nearly doubles by loading of Pt, which is an electron acceptor. Since Au loading blocks the $CH_3$· overoxidation pathway, more $CH_3$· species follow the coupling pathway to form $C_{2+}$ hydrocarbons.

To understand the origin of the high selectivity of Au-$CeO_2$-300 towards $C_{2+}$ hydrocarbons, ethane, the main product in methane oxidation was introduced as a reactant into the photocatalytic oxidation reaction over $CeO_2$ and Au-$CeO_2$-300 (Fig. 5). When $CeO_2$ is used as the photocatalyst, the main product is $CO_2$ at a yield of 49 $\mu$mol $h^{-1}$ (Fig. 5a). Apart from $CO_2$, $CH_4$ at a production rate of 15 $\mu$mol $h^{-1}$ is detected. Only a trace amount of $C_{2+}$ molecules (e.g., $C_2H_4$, $C_3H_8$, $C_3H_6$, $C_4H_{10}$ and $C_4H_8$) are observed in the product. In contrast, when Au-$CeO_2$-300 is used for ethane oxidation under identical conditions, the $CO_2$ product rate is only 5 $\mu$mol $h^{-1}$ (10 times lower than that over $CeO_2$), and $CH_4$ is not detectable in the products. The major product is switched to $C_4H_{10}$ at a high yield of

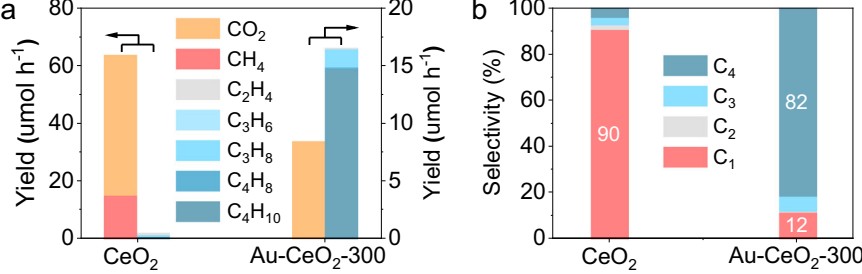

**Fig. 5 | Ethane oxidation reaction. a** Yield and (**b**) selectivity for ethane oxidation to different products over $CeO_2$ and Au-$CeO_2$-300. Reaction conditions: 50 mg catalyst, ethane to air = 5:1, GHSV = 120 000 mL $h^{-1}$ $g^{-1}$, Pressure = 1 bar, Temperature = 50 °C, 365 nm LED, light intensity = 100 mW $cm^{-2}$.

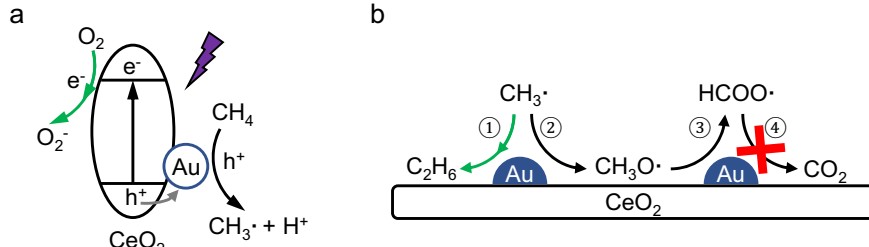

**Fig. 6 | Proposed methane oxidation process. a** Change separation-migration and reactant activation, and (**b**) the effect of Au on promoting C-C coupling, and suppressing overoxidation over Au-$CeO_2$−300.

15 µmol $h^{-1}$. To provide a clear product distribution for ethane oxidation over the photocatalysts, the selectivity of products is calculated based on the number of carbon in the molecules (Fig. 5b). It clearly shows that $C_1$ molecules with a high selectivity of 90% are the main products of photocatalytic ethane oxidation by $CeO_2$. In contrast, $C_4$ hydrocarbons account for 82% of the products when Au-$CeO_2$-300 is applied. To produce $C_1$ molecules, the C-C bond in ethane has to be broken. To form a $C_4$ molecule, a new C-C bond should be formed to connect two $C_2H_6$ molecules. Thus, the $CeO_2$ surface is prone to catalyse the C-C bond-breaking reaction. It further indicates that $C_2H_6$ could be overoxidised to $CO_2$ or transferred back to $CH_4$ even if it is formed by the coupling of $CH_3\cdot$ in photocatalytic methane oxidation over $CeO_2$. On the contrary, Au-$CeO_2$-300 can promote the C-C coupling reaction under photocatalytic conditions. Therefore, $CH_3\cdot$ radicals are more likely to couple into $C_2H_6$ on the Au-$CeO_2$-300 surfaces. This also explains the formation of $C_3$ and $C_4$ products by Au-$CeO_2$-300 in methane oxidation. It is worth noting that the ethane conversion rates over $CeO_2$ and Au-$CeO_2$-300 are 35 and 36 µmol $h^{-1}$, respectively, indicating that Au loading on $CeO_2$ has little effect on the conversion of ethane. Overall, these results provide strong evidence that Au on the surface of $CeO_2$ can promote the C-C bond coupling reaction and avoid overoxidation, thus boosting the selectivity towards $C_{2+}$ molecules in photocatalytic methane oxidation.

Based on the analysis above, a reaction mechanism for photocatalytic methane oxidation over Au-$CeO_2$-300 is proposed (Fig. 6). When the photocatalyst is irradiated by the 365 nm LED, electrons are populated to the CB of $CeO_2$, leaving holes on the VB of $CeO_2$. The photoelectrons reduce oxygen gas to produce superoxide radicals. The existence of $O_{VS}$ in $CeO_2$ improves the oxygen adsorption and reduction processes. Au acts as a hole acceptor and enhances charge separation efficiency. The photoholes at Au transferred from VB of $CeO_2$, then activate methane molecules to generate methyl radicals and protons (Fig. 6a). The protons are consumed by superoxide radicals, resulting in the formation of water. The coupling reaction of methyl radicals leads to the formation of ethane (step ① in Fig. 6b), which is the main pathway for the consumption of methyl radicals. This is due to the existence of Au nanoparticles, which boost the desorption and coupling of $CH_3\cdot$ radicals. A small portion of methyl radicals

undergoes overoxidation to $CH_3O\cdot$ and further to HCOO· (steps ② and ③ in Fig. 6b). On pristine $CeO_2$ surface, $CH_3\cdot$ is strongly bonded on the catalyst surface, which leads to severe overoxidation. Moreover, HCOO· is readily converted to $CO_2$, which further promotes overoxidation. Thus, a low selectivity towards $C_{2+}$ products is observed over $CeO_2$. In contrast, the presence of Au co-catalyst considerably reduces the efficiency of HCOO· oxidation (step ④ in Fig. 6), which limits the overoxidation process. Therefore, both a high yield and selectivity of $C_{2+}$ hydrocarbons are achieved in photocatalytic methane oxidation over Au-$CeO_2$-300.

In summary, this study reported the efficient and selective oxidative coupling of methane using Au-loaded $CeO_2$ by photon-phonon co-driven catalysis. The optimised Au-$CeO_2$-300 achieves a $C_2H_6$ yield of 755 µmol $h^{-1}$ (15,100 µmol $g^{-1}h^{-1}$) and a $C_{2+}$ selectivity of 98%, with the excellent stability of at least 120 h. The introduction of oxygen vacancies into Au-$CeO_2$ by pre-treatment at 300 °C further promotes the adsorption and activation of oxygen gas. Charge migration studies confirm that Au acts as a hole acceptor and improves charge separation in photocatalysis. Surface chemistry analysis shows that Au as a co-catalyst on $CeO_2$ can accelerate both the oxygen reduction and the methane oxidation half-reactions. In situ DRIFTS characterisation reveals that Au suppresses overoxidation by restraining the conversion of HCOO· radicals. Finally, the control experiment of the ethane oxidation reaction indicates that Au can promote the C-C coupling reaction and mitigate overoxidation, thereby boosting the selectivity towards $C_{2+}$ products during photocatalytic methane oxidation. These findings provide important insights into the catalyst design and fundamental understandings of catalytic partial oxidation of methane. With further light source engineering, reactor design and reaction optimisation, this study can offer a promising and potential solution for economical and sustainable methane conversion to high-value hydrocarbons under mild conditions.

## Methods
### Materials synthesis
Au was loaded on $CeO_2$ by a chemical reduction method. 50 mg $CeO_2$ (25 nm nanopowder, Sigma-Aldrich) was dispersed in 50 mL deionised

water and stirred for 30 min. Then, a certain amount of HAuCl$_4$·4H$_2$O (5 mg (Au) mL$^{-1}$, Sigma-Aldrich) solution was added and stirred for another 30 min. Subsequently, 5 mL NaBH$_4$ (2 mg mL$^{-1}$, Sigma-Aldrich) solution was added drop by drop in the above suspension. After stirring for 60 min, the product was washed by centrifugation and dried at 60 °C. To further improve the activity of the photocatalysts, Au-loaded CeO$_2$ was pretreated in a muffle furnace at various temperatures from 200 to 500 °C for 2 h.

### Characterisations

X-ray diffraction (XRD) was carried out with two Stoe STADI-P diffractometers, one equipped with a Mo Kα source ($λ = 0.7073$ Å, scanned from 3 to 35°) and another equipped with a Cu Kα source ($λ = 1.5418$ Å, scanned from 10 to 85°). Ultraviolet-visible diffuse reflectance spectroscopy (UV-Vis DRS) was measured by a Shimadzu UV-2550 spectrophotometer fitted with an integrating sphere. The reflectance values were directly converted to absorption by the Kubelka-Munk equation via the UV-Probe 2.33 software. X-ray photoelectron spectroscopy (XPS) was measured by a Thermo Scientific XPS instrument equipped with an Al Kα source ($hν = 1486.6$ eV). The in situ XPS measurement in dark and under light irradiation conditions was performed using the same sample. The XPS spectrum of the sample in dark was firstly measured. Then, another spectrum was acquired after the sample was irradiated by a 365 nm LED light for 10 min. Both spectra were calibrated using the C 1s spectra obtained in dark and light irradiation, respectively. The Au content in the photocatalysts was measured by inductively coupled plasma-atomic emission spectrometry (Agilent, ICP-OES 5800). Raman spectra were measured by a Renishaw InVia multi-channel Raman spectroscopy with a 514 nm excitation laser. Transmission electron microscopy (TEM) was carried out on a JEOL 2010 instrument. The size distribution of Au is obtained by measuring the diameter of >100 nanoparticles for each sample. Photoluminescence (PL) was collected from 330 nm to 800 nm by a Renishaw InVia spectroscopy with a 325 nm laser as the excitation source.

### Photocatalytic reaction test

The photocatalytic oxidative coupling of methane was carried out in a pressurised flow reaction system. The reactor used is a stainless steel cell equipped with a quartz window. All parts of the reaction system are connected by stainless steel tubings. The flow rates of methane, air and Ar were controlled by three mass flow controllers (Bronkhorst). Two pressure gauges were fitted before and after the reactor to monitor the pressure change during the photocatalytic reaction. No pressure drop was observed after the reaction. A regulator valve at the end of the reaction system was used to adjust and maintain the pressure in the reactor. A heating unit at the bottom of the catalyst bed is used to control the reaction temperature from room temperature to 200 °C. The photocatalyst was filtered onto a glass fibre membrane and fitted into the cell for the performance test. To prepare the membrane, 50 mg photocatalyst was dispersed in 200 mL deionised water and stirred for 30 min. Then, the suspension was filtered by the membrane and dried at 60 °C for 12 h. The membrane was fixed in the reactor by a stainless steel ring. The top and bottom parts of the reactor were sealed with a rubber ring and a stainless steel clamp. The light source used was a 365 nm LED (100 W, Beijing Perfect Light), a Xe lamp (300 W, Beijing Perfect Light) equipped with a 420 nm long-pass filter, and a 450 nm LED (100 W, Beijing Perfect Light). The light intensity of the 365 nm LED could be adjusted from 10 to 200 mW/cm$^2$, depending on the reaction conditions. The surface temperature of the catalyst was measured by a radiative thermometer (Extech Instruments, IR320). The product released from the regulator valve was directly connected to a GC (Varian 450) equipped with a TCD detector, a methanizer, and an FID detector. The isotopic labelling tests were performed in a quartz batch reactor (100 mL). The isotopic labelled

$^{13}$CH$_4$, CD$_4$ and $^{18}$O$_2$ are purchased from Sigma-Aldrich. After purging the reactor with N$_2$ (BOC) for 30 min, the feedstock (CH$_4$ + O$_2$, $^{13}$CH$_4$ + O$_2$, CD$_4$ + O$_2$, or CH$_4$ + $^{18}$O$_2$) with CH$_4$ to O$_2$ ratio of 10:1 was added into the reactor and irradiated by a 365 nm LED (100 W) for 15 min. The gas products were analysed by GC-MS (Shimadzu GCMS-QP2020 NX).

Methane conversion was calculated by:

$$Con._{CH_4} = \frac{nmuber\ of\ converted\ CH_4}{number\ of\ input\ CH_4} \times 100\% \quad (2)$$

Oxygen conversion was calculated by:

$$Con._{O_2} = \frac{nmuber\ of\ converted\ O_2}{number\ of\ input\ O_2} \times 100\% \quad (3)$$

The selectivities are calculated based on observable products as the following:

*Selectivity of C$_{2+}$*

$$= \frac{2 \times n_{C_2H_6} + 2 \times n_{C_2H_4} + 3 \times n_{C_3H_8} + 3 \times n_{C_3H_6} + 4 \times n_{C_4H_{10}}}{2 \times n_{C_2H_6} + 2 \times n_{C_2H_4} + 3 \times n_{C_3H_8} + 3 \times n_{C_3H_6} + 4 \times n_{C_4H_{10}} + n_{CO_2}} \times 100\%$$

$$(4)$$

*Selectivity of CO$_2$*

$$= \frac{n_{CO_2}}{2 \times n_{C_2H_6} + 2 \times n_{C_2H_4} + 3 \times n_{C_3H_8} + 3 \times n_{C_3H_6} + 4 \times n_{C_4H_{10}} + n_{CO_2}} \times 100\%$$

$$(5)$$

Carbon balance was calculated by:

$$Carbon\ balance = \frac{number\ of\ C\ in\ products}{number\ of\ converted\ CH_4} \times 100\% \quad (6)$$

Oxygen balance was calculated by:

Oxygen balance

$$= \frac{\frac{1}{2}nC_2H_6 + nC_2H_4 + nC_3H_8 + \frac{3}{2}nC_3H_6 + \frac{3}{2}nC_4H_{10} + nCO_2}{n\ of\ converted\ O_2} \times 100\%$$

$$(7)$$

The apparent quantum efficiency (AQE) was calculated based on the conversion of methane:

$$AQE = \frac{Number\ of\ electrons\ transferred}{Number\ of\ incident\ photons} \times 100\%$$

$$= \frac{\left(2 \times n_{C_2H_6} + 4 \times n_{C_2H_4} + 4 \times n_{C_3H_8} + 6 \times n_{C_3H_6} + 6 \times n_{C_4H_{10}} + 8 \times n_{CO_2}\right) \times N_A}{\frac{I \times A}{E_g \times J}} \times 100\% \quad (8)$$

Where $N_A$ is Avogadro's constant $6.02 \times 10^{23}$, I is the light intensity in the unit of mW cm$^{-2}$, A is the irradiation area 7 cm$^2$, $E_g$ is the energy of a photon with 365 nm wavelength and J is the amount of charge in one electron and used to transform the unit of photon energy from eV to J.

## Data availability

Data supporting this study are available in the Supplementary information and in the Source Data file provided with this manuscript. Source data are provided with this paper.

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

## Acknowledgements

J.T. and L.X. acknowledge the NSFC project (Grant No: 22250710677) and Beijing Municipal Project (C2022007). All authors also thank funding from UK EPSRC (EP/ S018204/2). C.W. and Y.X. acknowledge the UCL Dean's prize and China CSC scholarship.

## Author contributions

J.T. designed and supervised the overall project and oversaw all discussions. C.W. conducted the catalyst prepraration, most of the sample characterisations and activity tests. Y.X. assisted with the XRD, and photoelectrochemial measurements and discussion of results. L.X. assisted with the EPR measurement and data analysis. X.L. assisted with the activity test and mechanism discussion. E.C. assisted with the isotopic experiment. T.J.M. assisted with the in situ PIA and DRIFTS measurement. T.Z. performed the DFT simulations. Y.L. heavily contributed to the discussion of all experimental data. All authors contributed to the drafting of the manuscript and approved its final version.

## Competing interests

The authors declare no competing interests.
