## [Peer Review File · Nature Communications]

Selective oxidation of methane to C₂+ products over Au-CeO₂ by photon-phonon co-driven catalysisEditorial Note: This manuscript has been previously reviewed at another journal that is not operating a transparent peer review scheme. This document only contains reviewer comments and rebuttal letters for versions considered at *Nature Communications*.

REVIEWER COMMENTS

Reviewer #2 (Remarks to the Author):

The authors report that Au-CeO₂ photocatalytic oxidative coupling of methane is associated with high ethane formation rates and selectivity. Different from the photocatalysts reported in the past, this catalyst has a good visible light response, which is conducive to the efficient utilization of real sunlight. I suggest Nature communications to accept its publication with some minor revisions.

1. Despite some explanations made by the author, I still think that the author's quantification of by-product carbon dioxide is inaccurate, as shown in the chromatogram in Fig. S4. Perhaps the authors need to quantify the reaction products convincingly and accurately in future research work.
2. What is the temperature of the catalyst surface? Does this temperature have an effect on photocatalytic OCM?
3. I wonder why the author used such a low oxygen content in view of thermal catalytic OCM in oxygen concentration is higher (CH₄/O₂=5/1, 10.1002/anie. 202117201). The authors may consider evaluating the performance under the same conditions as the thermal catalytic system (600 mg catalyst, 10,000 mL gcat-1h-1, CH₄/O₂=5/1, no dilution) in future work.
4. Since Au-CeO₂ has a good visible light response, can the system achieve photocatalytic OCM under outdoor real sunlight irradiation?

We are very thankful to the reviewer for these constructive comments on our manuscript, which are very valuable and insightful for us to further improve the quality and scientific rigour of the manuscript. We have since carried out additional experiments and performance measurement. The manuscript and the Supporting Information have been revised accordingly and highlighted with a yellow background for easy referencing, as detailed below.

Reviewers' comments:

Reviewer #2 (Remarks to the Author):

The authors report that Au-CeO₂ photocatalytic oxidative coupling of methane is associated with high ethane formation rates and selectivity. Different from the photocatalysts reported in the past, this catalyst has a good visible light response, which is conducive to the efficient utilization of real sunlight. I suggest Nature communications to accept its publication with some minor revisions.

We thank the reviewer for pointing out the novelty and significance of the report and recognising the visible responsive nature and high performance of the Au-CeO₂ catalyst.

1. Despite some explanations made by the author, I still think that the author's quantification of by-product carbon dioxide is inaccurate, as shown in the chromatogram in Fig. S4. Perhaps the authors need to quantify the reaction products convincingly and accurately in future research work.

We thank the reviewer for this comment. Following this comment and to further confirm the amount of CO₂ produced, the product gas obtained under the optimised conditions was analysed by another GC with a separate gas line consisting of a methanizer and an FID detector. No valve switch was applied during the measurement to avoid noise background peaks. The result shows that the concentration of CO₂ (retention time = 4.2 min) in the product is 34 ppm (Fig. R1). This is very similar to the value in the previous measurement (30 ± 7 ppm). This further confirms that the method for CO₂ quantification is reliable.

Fig. R1 GC chromatogram showing the concentration of CO₂ produced by Au-CeO₂-300. Reaction conditions: 50 mg catalyst, methane to air = 200:1, GHSV = 480 000 mL h⁻¹ g⁻¹, Pressure = 5 bar, Temperature = 150 °C, 365 nm LED, light intensity = 200 mW cm⁻².

Fig. R1 is added to the revised Supporting Information as Fig. S4c.

2. What is the temperature of the catalyst surface? Does this temperature have an effect on photocatalytic OCM?

We thank the reviewer for the insightful comments. The surface temperature of the Au-CeO₂-300 under various conditions was measured using an infrared radiative thermometer (Extech Instruments, IR320). The results are summarised in Table R1. It can be seen that the surface temperature of the catalysts is increased from room temperature to 80-100 °C when the external heater is not used. Under the optimised reaction condition, the temperature of 150 °C was achieved by sum of the external heating and light-induced heating).

Table R1 Surface temperatures of Au-CeO₂-300 measured by an infrared radiative thermometer under reaction conditions with and without external heating.

Light source	Without external heating	Heating set to 150 °C
365 nm LED	83 °C	152 °C
450 nm LDE	95 °C	157 °C

To clarify the thermal catalysis only, the catalytic reaction at temperatures from 150 °C to 200 °C was performed in dark (Fig. R2). Only a trace amount of CO₂ (8 μmol h⁻¹) is observed when the reaction temperature reaches 200 °C. C₂₊ hydrocarbons are not detected.

Fig. R2 Product yield of Au-CeO₂ for catalytic methane conversion in dark at different temperatures. Reaction conditions: Reaction conditions: 50 mg catalyst, methane to air = 200:1, GHSV = 480 000 mL h⁻¹ g⁻¹, Pressure = 5 bar, dark.

On the other hand, without external heating and under 365 nm LED irradiation, the yield of C₂₊ is 317 μmol h⁻¹. The effect of reaction temperature on the catalytic performance suggests that high temperature results in the improved yield of all products (Fig. 1b). These results indicate that the selective photocatalytic conversion of methane into C₂₊ products is not primarily driven by heat while heating does play a secondary role. Therefore, coupling photocatalysis with thermal catalysis is beneficial to C₂₊ production (including both selectivity and yield).

Fig. 1b Product yield and C₂₊ selectivity of Au-CeO₂-300 tested at different reaction temperatures. Reaction conditions: 50 mg catalyst, methane to air = 200:1, GHSV = 480 000 mL h⁻¹ g⁻¹, Pressure = 5 bar, 365 nm LED, light intensity = 200 mW cm⁻². The reaction temperatures were measured by a thermal couple at the bottom of the catalyst bed.

Fig. R2 and Table R1 are added into the revised Supporting Information as Fig. S10 and Table S3, respectively.

The following description was added to Page 5 of the revised manuscript:

The product yield of Au-CeO₂-300 at higher temperatures from 150 to 200 °C was also measured in dark (Fig. S10). Only a trace amount of CO₂ is detected at 200 °C. Thus, it can be concluded that the selective catalytic conversion of methane into C₂₊ products is not driven by heat only while the catalytic performance of Au-CeO₂-300 at different temperatures (Fig. 1b) again suggests coupling photocatalysis with thermocatalysis can dramatically enhance the yield of C₂₊ products.

The following description is added to Page S28 of the revised Supporting Information.

Due to the strong absorption of Au-CeO₂-300 across the UV-Vis spectrum, the temperature of the catalyst surface may increase under irradiation due to the photothermal effect. Thus, the surface temperatures of Au-CeO₂-300 with and without external heating under various light sources were measured (Table S3). The surface temperatures range from 80 to 110 °C or from 150 to 160 °C under conditions without or with external heating, respectively.

3. I wonder why the author used such a low oxygen content in view of thermal catalytic OCM in oxygen concentration is higher (CH₄/O₂=5/1, 10.1002/anie. 202117201). The authors may consider evaluating the performance under the same conditions as the thermal catalytic system (600 mg catalyst, 10,000 mL gcat-1h-1, CH₄/O₂=5/1, no dilution) in future work.

We thank the reviewer for the constructive comments. The effect of oxygen content was investigated from pure CH₄ to CH₄ to air ratio ranging from 400:1 to 12:1 over Au-CeO₂-300, as shown in Fig. S8. When O₂ is absent in the reaction atmosphere, a low C₂H₆ yield of 71 μmol h⁻¹ is obtained. As the concentration of air in the reactant increases, the yields of all products are improved. This indicates that oxygen gas can have a positive effect on photocatalytic methane conversion. However, the yield of CO₂ surges from 30 to 124 μmol h⁻¹ when the methane-to-air ratio changes from 200:1 to 12:1 with the C₂H₆ yield slightly increased from 755 to 859 μmol h⁻¹. Overall, O₂ is indispensable in photocatalytic OCM, however, too much O₂ causes overoxidation of CH₄ to CO₂. To achieve both high yield and high selectivity of the high-value C₂₊ products, the methane-to-air ratio of 200:1 is chosen for subsequent studies.

Fig. S8 Product yield and C₂₊ selectivity of Au-CeO₂-300 tested under different methane-to-air ratios. Reaction conditions: 50 mg catalyst, GHSV = 480 000 mL h⁻¹ g⁻¹, Pressure = 5 bar, Temperature = 150 °C, 365 nm LED, light intensity = 200 mW cm⁻².

4. Since Au-CeO₂ has a good visible light response, can the system achieve photocatalytic OCM under outdoor real sunlight irradiation?

We thank the reviewer for this insightful comment. Due to safety risks caused by using high-pressure flammable methane outside research labs, it is very difficult to conduct photocatalytic methane oxidation under the actually solar irradiation using the current reaction system. To mimic the solar spectrum, the performance of Au-CeO₂-300 was measured under the irradiation of a Xe lamp equipped with an AM1.5 filter, which produces the same spectrum (Fig.R3) and intensity (100 mW cm⁻¹) as the solar irradiation. External heating was removed during the measurement. The surface temperature of the catalyst reached 90 °C during the measurement under simulated solar irradiation. The production rates of C₂H₆, C₂H₄, C₃H₈, and C₄H₁₀ are 104, 0.2, 4.5 and 51 μmol h⁻¹, respectively (Table R3). This performance is lower than that under visible light irradiation (λ > 420 nm) with external heating (150 °C). This further suggests that the coupling of photocatalysis with thermocatalysis is essential to achieve efficient oxidative coupling of methane. The C₂₊ selectivity obtained is 82%. This proves that the Au-CeO₂-300 is capable of driving the photocatalytic OCM reaction under simulated solar irradiation.

Fig. R3 Transmittance spectrum of AM 1.5 filter.

Table R3 Products yield over Au-CeO₂-300 under simulated solar irradiation. Reaction conditions: 50 mg catalyst, GHSV = 480 000 mL h⁻¹ g⁻¹, CH₄ to Air = 200:1, Pressure = 5 bar, No external heating, Xe lamp with an AM1.5 filter, light intensity = 100 mW cm⁻².

Product	C ₂ H ₆	C ₂ H ₄	C ₃ H ₈	C ₃ H ₆	C ₄ H ₁₀	CO ₂
Yield (μmol h ⁻¹)	104	0.2	4.5	0	0	51

The information in Table R3 is updated in Table S1 in the revised Supporting Information.

The following description was added to Page 5 of the revised manuscript.

The potential of Au-CeO₂-300 for oxidative coupling of methane under simulated solar irradiation was also measured using a Xe lamp equipped with an AM 1.5 filter (100 mW cm⁻²) without the external heating. The production rates of C₂H₆, C₂H₄, C₃H₈, and C₄H₁₀ are 104, 0.2, 4.5 and 51 μmol h⁻¹, respectively, suggesting the potential of the current system for solar driven methane conversion.

REVIEWERS' COMMENTS

Reviewer #2 (Remarks to the Author):

The authors have addressed all my comments. Thus, I'd like to recommend its publication in the current form.